# De Novo Assembly of the Polyhydroxybutyrate (PHB) Producer *Azohydromonas lata* Strain H1 Genome and Genomic Analysis of PHB Production Machinery

**DOI:** 10.3390/microorganisms13010137

**Published:** 2025-01-10

**Authors:** Daniele Traversa, Carlo Pazzani, Pietro D’Addabbo, Lucia Trisolini, Matteo Chiara, Marta Oliva, Angelo Marzella, Camilla Mandorino, Carla Calia, Guglielmina Chimienti, Caterina Manzari, Graziano Pesole, Maria Scrascia

**Affiliations:** 1Department of Biosciences, University of Milan, 20133 Milan, Italy; daniele.traversa@unimi.it (D.T.); matteo.chiara@unimi.it (M.C.); 2Department of Biosciences, Biotechnology and Environment, University of Bari Aldo Moro, 70125 Bari, Italy; carlo.pazzani@uniba.it (C.P.); pietro.daddabbo@uniba.it (P.D.); lucia.trisolini@uniba.it (L.T.); marta.oliva@uniba.it (M.O.); angelo.marzella@uniba.it (A.M.); camilla.mandorino@uniba.it (C.M.); carla.calia@uniba.it (C.C.); guglielminaalessandra.chimienti@uniba.it (G.C.); graziano.pesole@uniba.it (G.P.); 3Institute of Biomembranes, Bioenergetics and Molecular Biotechnology, Consiglio Nazionale delle Ricerche, 70126 Bari, Italy; caterina.manzari@uniba.it

**Keywords:** bioplastic, bacteria biopolymers, bioeconomy, polyhydroxyalkanoates, PHA, biodegradable

## Abstract

Polyhydroxybutyrate (PHB) is a biodegradable natural polymer produced by different prokaryotes as a valuable carbon and energy storage compound. Its biosynthesis pathway requires the sole expression of the *phaCAB* operon, although auxiliary genes play a role in controlling polymer accumulation, degradation, granule formation and stabilization. Due to its biodegradability, PHB is currently regarded as a promising alternative to synthetic plastics for industrial/biotechnological applications. *Azohydromonas lata* strain H1 has been reported to accumulate PHB by using simple, inexpensive carbon sources. Here, we present the first de novo genome assembly of the *A. lata* strain H1. The genome assembly is over 7.7 Mb in size, including a circular megaplasmid of approximately 456 Kbp. In addition to the *phaCAB* operon, single genes ascribable to PhaC and PhaA functions and auxiliary genes were also detected. A comparative genomic analysis of the available genomes of the genus *Azohydromonas* revealed the presence of *phaCAB* and auxiliary genes in all *Azohydromonas* species investigated, suggesting that the PHB production is a common feature of the genus. Based on sequence identity, we also suggest *A. australica* as the closest species to which the *phaCAB* operon of the strain H1, reported in 1998, is similar.

## 1. Introduction

The exponential increase in fossil-derived plastic waste and the growing demand for plastics [1,2] create the urgency to replace petrochemical-based plastics with biodegradable polymers [3,4,5]. Polyhydroxyalkanoates (PHAs) are a group of bio-based polyesters that are biodegradable, resemble synthetic plastics, produced by a range of diverse prokaryotes and accumulated in the cytoplasm as granules reserve of carbon and energy [6,7,8]. Based on the carbon atoms content per monomer unit, PHAs can be classified as short-chain-length (scl-PHAs), with 3–5 carbon atoms, and medium-chain-length (mcl-PHAs), containing 6–14 carbon atoms, per unit. Scl-PHAs are rigid and fragile with a high melting temperature and low glass transition temperature; they are the most abundant PHAs among prokaryotes. Mcl-PHAs are elastic with lower melting and glass transition temperatures, as compared to scl-PHAs [7].

Among the scl-PHAs, the homopolymer polyhydroxybutyrate (PHB) is one of those considered for large-scale production due to its biodegradability and biocompatibility, being proposed in medical and pharmaceutical fields as well [9]. Thus, PHB is currently regarded as one of the most promising PHAs for biotechnological applications, with increasing studies on PHBs producing bacteria and a growing interest to make PHB more competitive in commercial markets [10,11]. From the view of a circular economy, many efforts target efficient and low-cost PHB production by using, in addition to native producers, bacteria (e.g., *Escherichia coli*) engineered with heterologous PHB genes/operons with the objective of producing PHB efficiently from renewable biomass (e.g., whey waste, starch, wastewater) [12].

Gram-negatives, such as *Azohydromonas lata* (formerly *Alcaligenes latus*) strain H1 (DSM1123; ATCC 29714), *Azotobacter* spp., *Cupriavidus necator* (formerly *Ralstonia eutropha*) strain H16 (DSM 428; ATCC17699), *Pseudomonas* spp. and also recombinant *Escherichia coli* expressing the PHB biosynthetic genes from native producers strains, can accumulate large amounts of PHB and are considered the most promising systems for large-scale PHB production [9,13]. *C. necator* H16, in particular, represents the model organism for the study of PHB production [12,14]. Three main distinct pathways for PHB synthesis are described [7]; in *C. necator*, three genes, usually organized in an operon (*phaCAB*), are deemed sufficient for PHB biosynthesis: PHA synthase (*phaC*), 3-chetothiolase (*phaA*) and acetoacetyl-CoA reductase (*phaB*). PhaCs (the crucial function common to all the known pathways) are generally categorized into four classes (I to IV) based on amino acid sequence, in vivo substrate specificities of the enzymes and the number/composition subunits forming the catalytic complex [15]. An additional class (class V) was recently proposed by Tan and colleagues, based on the PhaC of *Janthinobacterium* sp. [16]. Beyond the three main genes (*phaA*, *phaB* and *phaC*), many auxiliary genes have been identified and characterized to encode important functions in controlling polymer accumulation and degradation. These include the regulatory gene *phaR*, the depolymerase *phaZ*, the extracellular oligomer hydrolase *phaY*, and the phasin *phaP* involved in granule formation/stabilization [17,18,19].

Since the availability of the reference *C. necator* H16 genome [20,21], the *phaCAB* operon, auxiliary genes associated with PHB production and isologs of PhaA (beta-ketothiolases) and PhaB (reductases) with different substrate specificity, have been characterized and studied in this bacterium [21,22]. However, equivalent considerations do not apply to the *A. lata* strain H1. Expanding the repertoire of known/characterized PHB genes could provide crucial insights for the optimization of biotechnological applications based on the genetic engineering of the pathway. Here, we present the first de novo genome assembly of *A. lata* strain H1, whose information might be exploitable for biotechnological applications [13,14]. Although the sequence of the *phaCAB* operon of *A. lata* H1 has been previously determined [23], we reasoned that the assembly of the genome sequence could offer significant advances in the identification of genes associated with PHB production and offer a more complete representation of the organization of the PHB operon, together with the number and configuration of auxiliary genes and potential isologs in *A. lata*. Moreover, our comparative genomic analyses of *phaCAB* and related genes within the genus *Azohydromonas* could advance the general knowledge of the genomic organization and conservation of these genes and inform the design of prospective biotechnological applications.

## 2. Materials and Methods

### 2.1. Genome Sequencing, Assembly and Annotation

*A. lata* strain H1 was obtained from the Leibniz institute DSMZ–German Collection of Microorganisms and cell culture GmbH (Braunschweig, Germany) (strain code DSM1123). The genomic DNA was isolated using the DNeasy Blood and Tissue Kit (Qiagen Inc., Hilden, Germany) Kit (QIAGEN, Hilden, Germany) and purified using Amicon Ultra-0.5. MWCO 30 kDa (Millipore, Burlington, MS, USA) according to the manufacturer’s protocols. The purity and concentration were assessed using a NanoDrop spectrophotometer and Qubit 3.0 (Thermo Scientific, Waltham, MA, USA), respectively. The library was prepared from the purified bacteria genomic DNA using Illumina DNA Prep (Illumina, San Diego, CA, USA). The sequencing was carried out using Miseq Reagent Kit v2 (500 cycles) and the Illumina MiSeq platform (Illumina, San Diego, CA, USA). Adaptor sequences and low-quality regions were removed by using Trimmomatic with default parameters [24]. Overlapping paired-end reads were merged by Pear [25]. Genome assembly was performed using the Unicycler workflow, using the SPAdes assembler ([26] GATK team). The following kmers sizes were used: 27, 53, 71, 87, 99, 111, 119, 127. Contigs shorter than 200 bp in size were discarded. The genome was deposited at NCBI under the GenBank accession number GCA_034427735.1. Gene annotation was performed by the NCBI Prokaryotic Genome Annotation Pipeline (PGAP) [27]. The annotation of rRNA were manually refined by performing blastn (v2.9.0-2) [28] sequence similarity searches and manual alignment with the 5S, 16S and 23S rRNA genes from the draft genome assembly of the *A. lata* type strain NBRC 102462 (GenBank accession number GCA_001571085.1). CRISPR-Cas systems and CRISPR array(s) were annotated by CRISPRCasFinder with default parameters [29]. The potential self-targeting of spacers was assessed by blastn sequence similarity searches of the spacers’ sequences in the genome. Default parameters were used and only matches showing an e-value ≤ 10^−50^ and identity ≥ 90% were considered. The spacers’ sequences were also searched in the NCBI nr/nt (https://ftp.ncbi.nlm.nih.gov/blast/db/ accessed on 4 March 2024) database to assess the presence of sequences of exogenous origin. Default parameters were used. The CRISPR array subtype was inferred based on sequence similarity with the most similar sequence in the NCBI nr/nt database (https://ftp.ncbi.nlm.nih.gov/blast/db/ accessed on 6 March 2024). Candidate transposases, resistance genes and Toxin–Antitoxin (TA) systems were inferred according to the PGAP annotation. The general features and genome statistics were computed using custom scripts written in Python (v 3.8.10) and R (v 4.2.0) and standard bash shell utilities. The circular map of the candidate plasmid sequence was generated with the web-based implementation of Proksee [30].

### 2.2. ANI and dDDH

The Average Nucleotide Identity (ANI) and Digital DNA–DNA hybridization (dDDH) were computed by considering the complete collection of publicly available *Azohydromonas* genome assemblies accessible through NCBI Refseq (https://www.ncbi.nlm.nih.gov/assembly/?term=Azohydromonas accessed on 18 December 2023) (Appendix A). The ANI values were computed using OrthoAniU [31]. The dDDH values were determined using the Genome-to-Genome Distance Calculator (GGDC-3.0) with the recommended settings [32].

### 2.3. PHB Genes Detection and Class Designation of PhaCs

PHB genes were annotated using blastp sequence similarity searchers with PHB genes from *C. necator* H16 (GeneBank AM260479.1) [21]. Only blastp matches showing an e-value ≤ 10^−50^ and identity ≥ 40% were considered. Candidate PhaC protein sequences from *A. lata* H1 were assigned to one of the five classes of PhaC enzymes based on similarity, as determined by blastp searches (with default parameters), with respect to an arbitrarily selected collection of representative sequence for every class (Appendix A). Only matches with an e-value ≤ 10^−50^ were considered, and the class was assigned based on the best blastp match. Candidate PhaC protein sequences were aligned by using the online version of Clustal Omega [33], and the alignment was visualized with JalViewer version 2.11.4.1 [34].

## 3. Results

### 3.1. Genome Assembly of A. lata H1

The genome assembly of *A. lata* strain H1 is accessible at GenBank under the accession number GCA_034427735.1. The draft genome is over 7.7 Mb in size and includes a circular megaplasmid of approximately 456 Kbp (Figure 1 and GenBank: JAXOJX010000001.1).

The genome assembly consists of 302 scaffolds and the N50 value is 89,247 bp. The descriptive statistics are reported in Table 1.

The G + C content is 65.6% for the megaplasmid and 68.7% for the remaining scaffolds. A total of 1 rRNA 5S, 3 rRNA 16S, 2 rRNA 23S, and 56 tRNA genes were identified; none of these are localized in the megaplasmid. The PGAP annotation predicted 6945 protein coding genes, of which 408 were on the megaplasmid; a putative function was assigned to 5933 proteins (megaplasmid 300), while 14.5% of the proteins (1012) were annotated as “hypothetical proteins”. A total of 49 genes encoding putative transposases were annotated by PGAP; 16 transposases were localized on the megaplasmid and 10 formed a potential cluster spanning approximately 100 Kb in size (Figure 1 and Figure 2). Based on the PGAP annotation, 35 transposases were assigned to eight different families while 14 were not assigned to a family (Table 2 and Appendix A).

TA systems play a role in maintaining plasmids in bacteria cells [35]. They usually consist of two proteins (antitoxin and toxin) encoded by two juxtaposed genes, organized in an operon with the antitoxin gene more frequently upstream than the toxin gene [36]. TA operons characterized by the toxin gene upstream of the antitoxin have been recently reported [37]. Based on the nature (protein or RNA) and mode of action of antitoxins, TA systems have been classified into different classes [36], with type II systems being the most abundant and extensively studied. Candidate TA systems were manually searched in the DSM1123 megaplasmid inspecting candidate toxin and antitoxin genes, according to the PGAP annotation. Two distinct putative type II TAs were identified in the *A. lata* megaplasmid (Table 3).

The first system (system II-A hereafter) belongs to the RelE/ParE family and is composed of two protein coding genes: the toxin (locus tag SM757_01580) and the HigA antitoxin (locus tag SM757_01585). The second system (system II-B hereafter) has a similar configuration, being formed by the RelE/ParE toxin (locus tag SM757_01815) and HigA antitoxin (locus tag SM757_01820). System II-B includes an additional gene (locus tag SM757_01810) encoding for the VapB antitoxin. Whereas three components of type II TA systems have been previously described ([36], Toxin-Antitoxin Database. http://bioinfo-mml.sjtu.edu.cn/TADB/, accessed on 19 April 2024), to the best of our knowledge, this is the first reported instance of a potential VapB/RelE/HigA TA system. Interestingly, 10 Kbp upstream of system II-B, the presence of a predicted antitoxin AbiEi domain protein (locus tag SM757_01845), which could represent a partial abortive type IV system (AbiEi/AbiEii) [38], was also noticed. All the candidate TAs co-localize in the same plasmid region adjacent to the cluster of the 10 predicted transposase genes described above (Figure 1).

### 3.2. Nucleotide Identity Levels in the Genus Azohydromonas

The current taxonomic classification of *A. lata* collocates the species in the family Alcaligenaceae, order Burkholderiales. Recently, Mogro and colleagues have published a phylogeny of the genus supporting the classification of Azohydromonas in the family Comamonadaceae [39]. The phenetic clustering of nucleotide identity levels among Azohydromonas genome assemblies, as available in December 2023 (Appendix A), indicates the reference strain *A. lata* NBRC102462 as the most similar sequence to *A. lata* H1 (Figure 3), with an ANI of 98.7%, which is well within the range observed between isolates of the same species [40]. At the species level, *A. aeria* (84.5), *A. australica* (85.7), *A. caseinilytica* (84.6) and an uncultured *Azohydromonas* specimen (84.8) had the highest level of sequence identity with *A. lata*. Additionally, nearly identical patterns of genome identity levels were recovered when similarity metrics based on dDDH [41] were considered (Figure 4).

Further, our ANI and dDDH analyses suggest high levels of sequence identity between *A. sediminis* (strains YIM 73032 and SYSU G00088) and *Calidifontimicrobium* sp. strain SYSU G02091, but a relatively low (ANI~76%) level of sequence identity between these species and the other *Azohydromonas* species. In consideration of the fact that genome sequence identity levels ranging between 72 and 75% are normally used to delineate a bacterial genus [40], these patterns of genome identity levels—along with the observation that *A. sediminis* strains present considerably smaller genomes compared with other *Azohydromonas* (average size 3.6 Mb with respect to 7.2 Mb, Appendix A)—might advocate for the reclassification of *Calidifontimicrobium* sp. strain SYSU G02091 under the genus *Azohydromonas*, or eventually suggest the classification of the three strains (YIM 73032, SYSU G00088 and SYSU G02091) under a genus different from *Azohydromonas*.

### 3.3. CRISPR-Cas Systems

Soil was the environment from which *A. lata* H1 has been isolated. In this complex environment, exogenous genetic elements, such as plasmids and phages, may represent a life threat (e.g., phages) or a metabolic burden (e.g., plasmids). The invasion of such elements is counteracted by the adaptive immunity defense mechanism CRISPR–Cas system. Studying the presence/absence of CRISPR–Cas systems in the genome allows us to better delineate the strain adaptation in its natural habitat based on the balance between the protection provided by CRISPR systems and their possible deleterious effects (e.g., self-targeting spacers), the role played by exogenous genetic elements (e.g., plasmids, phages, etc.) in strain evolution and the horizontal transfer of the CRISPR system. A CRISPR array was identified in the draft assembly of the *A. lata* H1 genome (contig JAXOJX000000044 25075-26262) by CRISPRCasFinder [29]. However, no cas genes were annotated by PGAP. The identified CRISPR array contained 20 direct repeats (DRs) with the consensus DR sequence GTATTTCCCGCGCGAGCGGGGATAAACCG showing the highest level of similarity with putative subtype I-E DRs in the genome assembly of *Cronobacter sakazakii* (GenBank accession number GCA_009648895.1). Sequence similarity searches throughout the genome of *A. lata* H1 did not indicate any potential self-targeting spacer. The candidate spacer sequences did not show any similar detectable protospacers with publicly available plasmids and/or phage sequences, suggesting that the spacers’ protospacers might not have been sequenced. Although we cannot exclude that the lack of a set of cas genes in the genome might result from incomplete assembly or inaccurate annotation, the observations reported above might be compatible with an exogenous origin of the identified CRISPR array. Alternatively, a CRISPR array not associated with known cas genes might indicate a possible role of the array beyond the adaptive immunity (e.g., regulatory function) [42] or association with Cas proteins yet to be identified.

### 3.4. Resistance Proteins

As stated in the previous section, *A. lata* H1 was purified from soil samples collected in Berkeley University (California) before September 1977. Soil bacteria might have been exposed to a variety of environmental *stimuli*, including competition with other microbial populations and/or the exposure to chemical pollutants. Hence, the identification of candidate resistance genes might be an indication of the adaptation of the *A. lata* H1 to competitive habitats. A total of 11 ORFs possibly associated with resistance proteins were annotated by PGAP (Table 4).

None of these are localized on the megaplasmid. These candidate resistance genes can be broadly categorized into six different classes: heavy metals resistance (arsenical-cadmium (SM757_03940), cobalt–zinc–cadmium (SM757_03785), toxic metal(loid) resistances (four genes tellurite TerB: SM757_05525, SM757_19280, SM757_24300, SM757_24860), toxic compounds resistance (chromate resistance, three genes: SM757_03450, SM757_03465, SM757_05070), glyoxalase/bleomycin/dioxygenase resistance (SM757_12920), and organic hydroperoxide resistance (SM757_22515). Although, in the absence of an experimental validation, predicted patterns of resistance cannot be confirmed, we speculate that the presence of genomic regions encoding for six different families/superfamilies of resistance genes suggests a possible wide resilience of *A. lata* H1 to different environments. This is consistent with the environmental origin of the strain.

### 3.5. Annotation of PHB Related Genes

The ability of the *A. lata* strain H1 to produce PHB is an attractive feature for biotechnological applications. The cloning and molecular analysis of the strain’s operon *phaCAB* has been previously reported by Choi et al. [23]. Here, we report a detailed description of the *phaCAB* operon and auxiliary genes related to polyhydroxybutyrate production in the genome of *A. lata* H1. The model organism *C. necator* H16 was used as a query to identify candidate orthologous genes in the proteome of *A. lata* H1. A total of 27 PHB-related genes were identified (Table 5 and Appendix A). Of these, four were annotated as *phaC*, one as *phaB* and six as *phaA*. An arrangement of genes compatible with a *phaCAB* operon was identified at genomic coordinates JAXOJX010000042 5833-9548 as follows: SM757_22330, poly(R)-hydroxyalkanoic acid synthase (*phaC*) (1611 pb); SM757_22335, acetyl-CoA acetyltransferase (*phaA*) (1179 bp); and SM757_22340 acetoacetyl-CoA reductase (*phaB*) (738 bp); three extra *phaCs* (named *phaC1* and *phaC2* in coherence with *C. necator* H16 annotation) and five extra *phaA* (SM757_00745 on the megaplasmid) were predicted. These extra PHA genes did not show an operon architecture (e.g., *phaCA*), as previously described in other species [18,43]. The PHA synthase encoded by the *phaC1* gene in the *phaCAB* operon reported for *C. necator* H16 and *A. lata* H1 was already classified as class I [15]. To establish the class of the predicted extra PhaCs, the protein sequence similarity, with respect to a selection of representative protein from class I to V PhaCs [16,44], was used (Appendix A). Based on this analysis, the *A. lata* PhaC1s (SM757_22330 and SM757_05625) are putatively assigned to class I while the PhaC2s (SM757_29010 and SM757_26710) were assigned to class V (Figure 5, Appendix A).

Rehm and colleagues [15] identified eight key amino acid residues (S260, C319, G322, D351, W425, D480, G507 and H508) by referring to *C. necator* H16 PhaC1 that are universally conserved in all PHA synthases, suggesting an important role of those residues in protein function. The conservation of these amino acids was more recently confirmed by Tan and colleagues in Class V PhaCs as well [16]. We performed an equivalent analysis for both the *C. necator* H16 PhaC2 and the extra candidate PhaCs annotated in *A. lata* H1. All the eight key residues, including the putative catalytic triad (C319, D480 and H508 based on *C. necator* H16 PhaC1), were conserved (Figure 6 and Appendix A). The isologs of PhaA (beta-ketothiolases) and PhaB (reductases) with different substrate specificity have been previously reported in *C. necator* H16 [21,22]. Based on the PGAP annotation, seven candidate PhaA isologs (thiolases) were identified in *A. lata* H1, and among these, one (SM757_01140) is encoded on the megaplasmid (Table 5 and Appendix A). Based on the same criteria, three candidate PhaB isologs (reductase) have been predicted (ORFs SM757_08215, SM757_31855 and SM757_23050). Interestingly, our sequence similarity analyses identified, in the genome of *A. lata* H1, at least one candidate homolog for all of the auxiliary PHA genes described in the model *C. necator* H16, including the following: two phaY hydrolases (locus tags SM757_26695 and SM757_23895); two polyhydroxyalkanoate depolymerase phaZ (locus tags SM757_08600 and SM757_15980); one phaP phasin family protein (locus tag SM757_26850) and a phaR transcriptional regulator (locus tag SM757_25825).

### 3.6. PhaCAB and Auxiliary Genes in the Azohydromonas Genus

The same procedure used for the annotation of candidate PHA genes in *Azohydromas lata* H1, was applied to the complete collection of publicly available *Azohydromonas* spp. genome assemblies (Appendix A). Candidate *phaA*, *phaB* and *phaC* genes, as well as auxiliary PHA genes and putative isologs, were recovered in all the genomes considered. Potentially intact *phaCAB* operons were observed in the following: *A. lata* NBRC 102462, *Calidifontimicrobium* sp. SYSU G02091, *A. sediminis* (SYSU G00088 and YIM 73032), *A. caseinilytica* G-1-1-1-4, whereas in *Azohydromonas aeria*, *Azohydromonas australica* and the uncultured *Azohydromoas* sp., a partial *phaCA* operon was identified. In these last three cases, however, both genes were consistently localized at one of the extremities of a contig in all the genome assemblies, potentially suggesting possible issues in the assembly due to multiple copies of the operon. In line with this hypothesis, a complete candidate *phaB* gene was annotated in *A. aeria* and *A. australica* in a non-adjacent region with respect to the *phaCA* incomplete operon. Further, we observed that *A. aeria* and *A. australica* were also associated with an increased number of candidate PHB genes, auxiliary genes and isologs (Appendix A). Conversely, and probably due to their reduced genome size, *Calidifontimicrobium* sp. SYSU G02091 and *A. sediminis* strains SYSU G00088 and YIM 73032 displayed a more limited repertoire of candidate PHB related genes and isologs. The sequence of the *phaC*, *phaA* and *phaB* from the *phaCAB* operon, both annotated by our analyses in the genome of *A. lata* H1 and reported by Choi et al. [23], were independently compared with candidate *phaCAB* genes from other *Azohydromonas* spp. As expected, high levels of similarity were observed between the *A. lata* H1 and the *A. lata* type strain NBRC102462 genes (range 99.1–99.8%) while the sequences reconstructed by Choi had the highest levels of similarity (range 94.3–100%) with *Azohydromonas australica* DSM 1124 *phaCAB* genes (Table 6, Table 7 and Table 8). These results indicate that the *phaCAB* operon described by Choi et al. was probably isolated from *A. australica*. It might be explained in the light of the recent reclassification/split about the two *Alcaligenes latus* isolates IAM 12599T and IAM 12664 into two distinct species: *A. lata* and *A. australica* [45]. 

## 4. Discussion

The increasing usage of plastic in many anthropic activities has caused a global environmental crisis of plastic pollution, with serious risks for animal and human health. PHB is a bio-based biologically degradable polymer suitable for producing biodegradable plastic, representing an eco-friendly alternative to synthetic plastic [46]. Being nontoxic, it also has a promising role in designating strategies related to regenerative medicine and tissue engineering [9]. The improved understanding of both PHB metabolism (e.g., synthesis and degradation) and the genetic repertoire in native and recombinant bacteria producers, might contribute to a widespread industrial application of PHB-based materials to meet the growing global demand of green bioplastics from renewable sources [46]. Here, we report the first draft genome assembly of the bacterium *Azohydromonas lata* H1, a PHB producing strain with potential biotechnological applications.

The taxonomic assignment of *A. lata* H1 within the genus *Azohydromonas*, was confirmed by a sequence similarity matrix based on ANI and dDDH. However, patterns of genome sequence identity, coupled with the observation of substantial differences in the size of the genome, might advocate for a partial revision of the genus *Azohydromonas* itself. More specifically, *A. sediminis* (YIM 73032, SYSU G00088) and *Calidifontimicrobium* sp. (SYSU G02091) presented levels of ANI of around 76% with all the other *Azohydromonas* species, a value that is considered borderline for the delineation of a bacterial genus; moreover, the size of the genome was significantly reduced in these species (~3.8 Mb compared to ~7.4 Mb of other *Azohydromonas* species, Appendix A). Further, since there is a high level of identities between the sequences of *A. sediminis* (YIM 73032, SYSU G00088) and *Calidifontimicrobium* sp. (SYSU G02091), they should be reclassified under the same species.

An analysis of the genome sequencing also revealed the presence of a 456 Kbp megaplasmid, not reported in the genome assembly of the reference strain *A. lata* NBRC 102462. The megaplasmid harbors two distinct type II TA systems (here named II-A and II-B). The II-B system is, to the best of our knowledge, the first report of a VapB/RelE/HigA three component system.

By focusing on genes implicated in PHB production, in addition to the identification of the *phaCAB* operon, different auxiliary genes associated with PHB utilization and granule formation were detected. These included *phaR*, *phaP*, *phaY*, as well as extra copies of *phaA* and *phaC* and a number of potential *phaA* and *phaB* isologs. Based on sequence similarity, four distinct PhaC functions (two PhaC1 and two PhaC2 based on *C. necator* H16 annotation) were found, of which the two PhaC1 were putatively assigned to class I while the two PhaC2 were assigned to class V. This finding is consistent with the possibility of a broad substrate utilization for PHB production, although experimental investigations are required to validate such a hypothesis. Our comparative genome analyses indicate the presence of PHB and its related genes in all the *Azohydromonas* genome assemblies considered, suggesting that all the considered species of the genus *Azohydromonas* are endowed with the molecular machinery for PHB production. According to the observed patterns of PHB gene distribution, *A. australica* is the species with the largest repertoire of PHB genes within the genus *Azohydromonas*, and speculations purely based on gene dosage/gene number would suggest high levels of PHB production in this species. Consistent with this hypothesis and in consideration of our sequence similarity analyses, we suggest that the *phaCAB* described by Choi et al. in 1998 [23]—which was isolated from a specimen with “high concentration with high productivity” of PHB—is assigned to *A. australica* and not *A. lata* H1.

By using modern approaches based on genome sequence and identity/similarity metrics, we derive a more precise and unequivocal identification of the species of origin of the *phaCAB* operon originally described by Choi et al. in 1998 [23], and we speculate that—at least in *A. australica*—increased PHB production might depend on PHB gene number/gene dosage, rather than on the optimization of the catalytic activity of a specific enzyme. This observation, coupled with the widespread distribution of PHB-related genes across the genus *Azohydromonas*, as evidenced by our analyses, prompts for further functional studies for a more accurate characterization of the levels of PHB production, underlying molecular pathways and potential biotechnological applications of *Azohydromonas*.

## 5. Conclusions

Since the elevated production cost of PHB, compared to petroleum-based plastics, is still a significant barrier to the wide use of this biopolymer in industrial settings, it might be crucial to use an omics approach (genomics, proteomics and transcriptomics) in studies focusing on bacteria PHB accumulation by utilizing organic low-cost wastes as metabolic substrates at different growth conditions [10]. The results of our study, in addition to the first draft genome sequence of *A. lata* strain H1, supplies a comprehensive delineation of the genetic repertoire of PHB genes in the genus *Azohydromonas* and underlines the importance of comparative genomics for informing the design of biotechnological applications based on microbial species. The genome sequencing of PHB producers, such as *A. lata*, helps to increase knowledge of the genetic network involved in PHB biosynthesis. Our study suggests that the PHB pathway is a common evolutionary feature of the genus *Azohydromonas* and provides data for further analysis on the ecological and genomic issues of PHB production. This knowledge can be exploited from a biotechnological point of view and for studies (with an omics approach) on PHB production, evolutionary dynamics of PHA operons and auxiliary genes and their possible horizontal gene transfer.

## Figures and Tables

**Figure 1 microorganisms-13-00137-f001:**
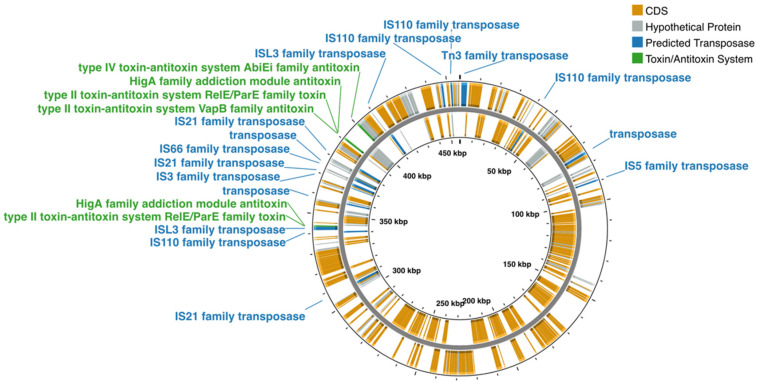
Circular map of the candidate plasmid generated with the web-based implementation of Proksee. Orange: CDS univocally annotated by PGAP; gray: CDS without a predicted function; blue: predicted transposases; green: TA system-inferred proteins.

**Figure 2 microorganisms-13-00137-f002:**
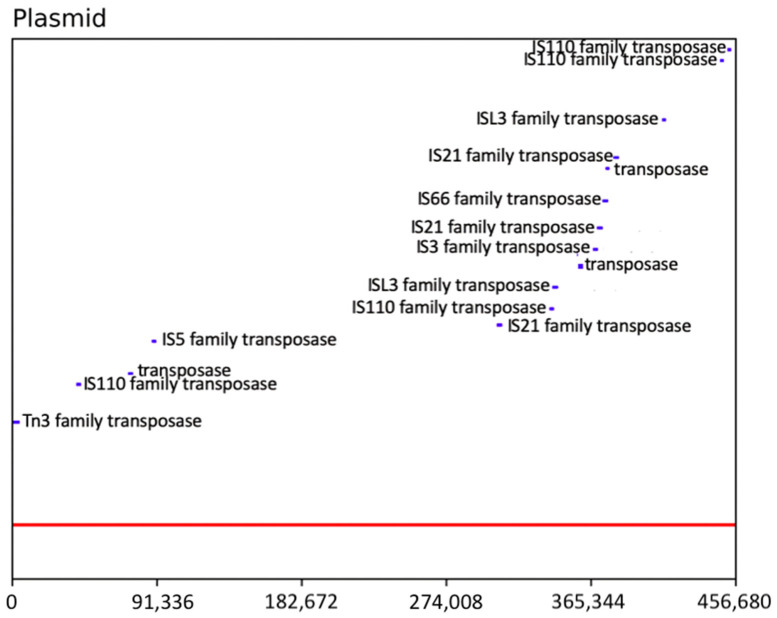
Linear representation of the transposes localized on the megaplasmid. The red line corresponds to the plasmid reported in a linearized fashion while the blue bars spanning above are the projections of the predicted transposases on the plasmid. The potential cluster of transposases is visible in the last quarter (from 310 kbp to 410 kbp) of the plasmid from the IS21 family transposase (left side) to the ISL3 family transposase (right side).

**Figure 3 microorganisms-13-00137-f003:**
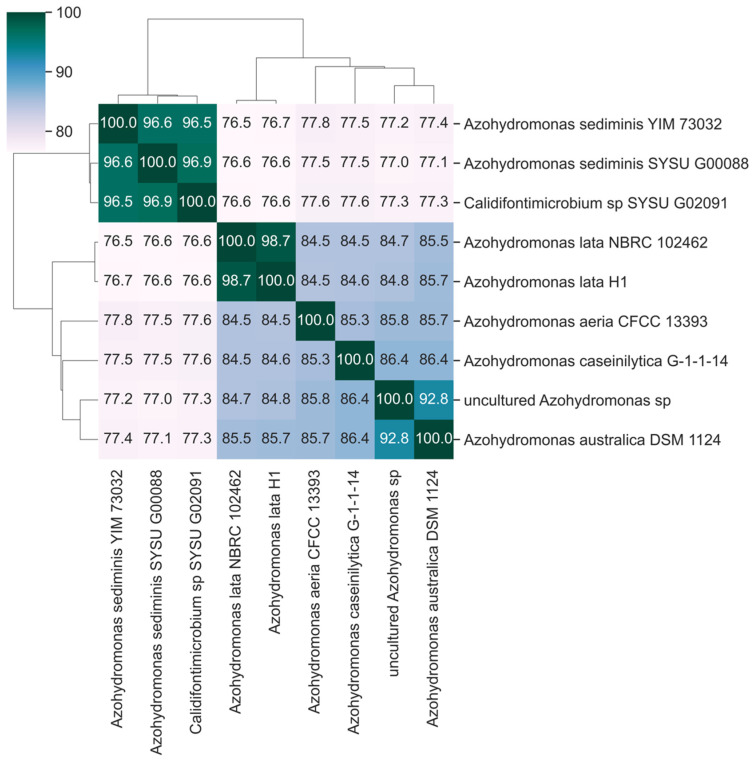
Heatmap of the ANI values. The heatmap is calculated by OrthoAniU between each pair of *Azohydromonas* isolates assemblies, including self-comparison (main diagonal). Darker shades of the color indicate higher ANI values. Accession numbers are listed in Appendix A.

**Figure 4 microorganisms-13-00137-f004:**
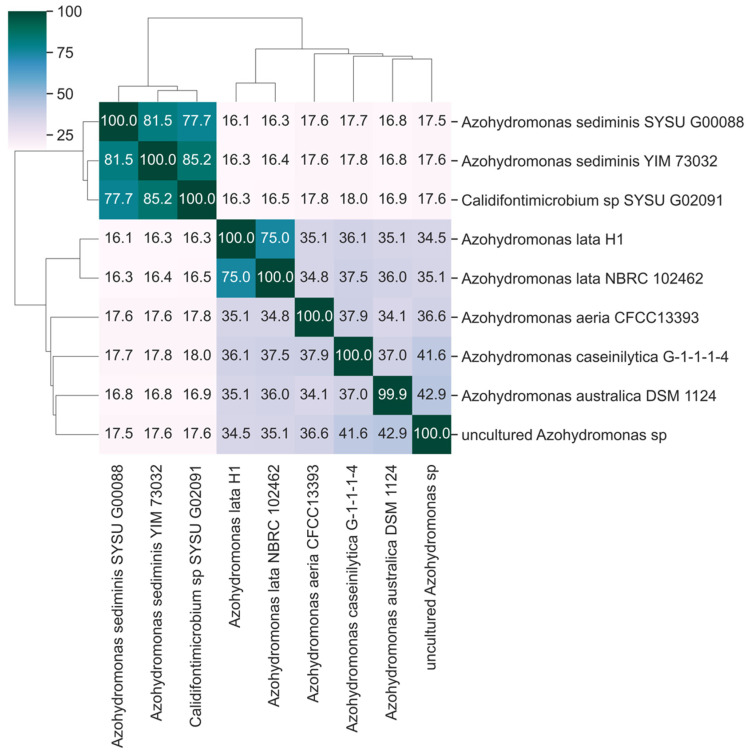
Heatmap of the dDDH values. The heatmap has been computed using the Genome-to-Genome Distance Calculator (GGDC-3.0) with recommended settings between each pair of *Azohydromonas* isolates assemblies. Self-comparison is on the main diagonal and a darker color suggests higher dDDH values. Accession numbers are listed in Appendix A.

**Figure 5 microorganisms-13-00137-f005:**
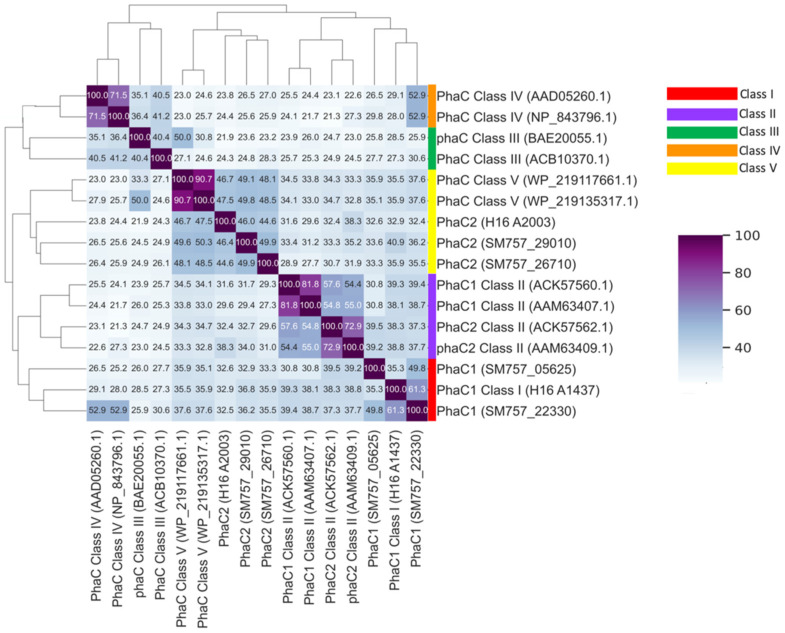
Heatmap of blastp sequence similarity analysis among collection of representative PhaCs classes (I–V). Darker shades of the color mark higher identity values. Self-comparison occupies the main diagonal.

**Figure 6 microorganisms-13-00137-f006:**
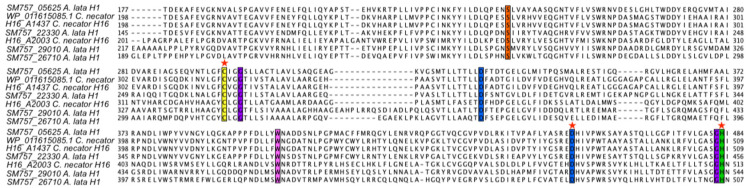
Portion of the MSA of candidate PhaC proteins computed by the online version of Clustal Omega and visualized with JalViewer. Colored columns are the 8 conserved residues. Residues of the catalytic triad are highlighted with red stars on the top of the corresponding column.

**Table 1 microorganisms-13-00137-t001:** General features of the genome.

Feature	Chromosome(s)	Megaplasmid
Size (bp)	7,328,099	456,680
G + C ratio (%)	68.7	65.6
Percentage coding	88.24	81.41
tRNA	56	0
rRNA 5S, 16S, 23S	1, 3, 2	0
Transposases	33	16
Total number of CDSs	6537	408
No. of CDSs with assigned function	5633	300
CDSs with unknown function	904	108

**Table 2 microorganisms-13-00137-t002:** Predicted transposases.

Transposase Family	Chromosome(s)	Plasmid
Tn3	2	1
IS110	1	4
Transposase	11	3
IS5	1	1
IS21	1	3
ISL3	3	2
IS3	1	1
IS66	6	1
IS630	7	0

**Table 3 microorganisms-13-00137-t003:** TA systems predicted to be on the megaplasmid.

Locus Tag	Product
SM757_01580 *	type II TA system RelE/ParE toxin
SM757_01585 *	HigA addiction module antitoxin
SM757_01810 **	type II TA system VapB antitoxin
SM757_01815 **	type II TA system RelE/ParE toxin
SM757_01820 **	HigA addiction module antitoxin
SM757_01845	type IV TA system AbiEi antitoxin

* system II-A. ** system II-B.

**Table 4 microorganisms-13-00137-t004:** Regions putatively encoding for resistance proteins.

Locus Tag	Accession of Contig	Product
SM757_03450	JAXOJX010000003.1	chromate resistance protein
SM757_03465	JAXOJX010000003.1	chromate resistance protein
SM757_03785	JAXOJX010000003.1	cobalt–zinc–cadmium resistance protein
SM757_03940	JAXOJX010000003.1	ArsI/CadI family heavy metal resistance metalloenzyme
SM757_05070	JAXOJX010000004.1	chromate resistance protein
SM757_05525	JAXOJX010000005.1	TerB tellurite resistance protein
SM757_12920	JAXOJX010000019.1	glyoxalase/bleomycin resistance/dioxygenase protein
SM757_19280	JAXOJX010000033.1	TerB tellurite resistance protein
SM757_22515	JAXOJX010000042.1	organic hydroperoxide resistance protein
SM757_24300	JAXOJX010000049.1	TerB tellurite resistance protein
SM757_24860	JAXOJX010000051.1	TerB tellurite resistance protein

**Table 5 microorganisms-13-00137-t005:** PHA-related functions predicted in the genome.

ALH1 Locus Tag	ORF Length	CONTIG	ALH1 Predicted Function	Orthologous		
				Locus Tag	Gene(s)	Function
SM757_22330	1611	JAXOJX010000042 *	class I poly(R)-hydroxyalkanoic acid synthase	H16_A1437	phaC1	Poly(3-hydroxybutyrate) polymerase
SM757_05625	1752	JAXOJX010000005	class I poly(R)-hydroxyalkanoic acid synthase	H16_A1437	phaC1	Poly(3-hydroxybutyrate) polymerase
SM757_29010	1884	JAXOJX010000076	poly-beta-hydroxybutyrate polymerase N-terminal domain-containing protein	H16_A2003	phaC2	Poly(3-hydroxybutyrate) polymerase
SM757_26710	1767	JAXOJX010000060	alpha/beta fold hydrolase	H16_A2003	phaC2	Poly(3-hydroxybutyrate) polymerase
SM757_22335	1179	JAXOJX010000042 *	acetyl-CoA C-acetyltransferase	H16_A1438	phaA	Acetyl-CoA acetyltransferase
SM757_28485	1185	JAXOJX010000072	beta-ketothiolase BktB	H16_A1438	phaA	Acetyl-CoA acetyltransferase
SM757_00745	1185	JAXOJX010000001	acetyl-CoA C-acyltransferase	H16_A1438	phaA	Acetyl-CoA acetyltransferase
SM757_27765	1209	JAXOJX010000067	acetyl-CoA C-acyltransferase	H16_A1438	phaA	Acetyl-CoA acetyltransferase
SM757_19820	1179	JAXOJX010000035	acetyl-CoA C-acyltransferase	H16_A1438	phaA	Acetyl-CoA acetyltransferase
SM757_22610	1185	JAXOJX010000043	acetyl-CoA C-acyltransferase	H16_A1438	phaA	Acetyl-CoA acetyltransferase
SM757_24435	1215	JAXOJX010000049	3-oxoadipyl-CoA thiolase	H16_A1438	phaA	Acetyl-CoA acetyltransferase
SM757_07035	1206	JAXOJX010000007	3-oxoadipyl-CoA thiolase	H16_A1438	phaA	Acetyl-CoA acetyltransferase
SM757_09505	1206	JAXOJX010000012	3-oxoadipyl-CoA thiolase	H16_A1438	phaA	Acetyl-CoA acetyltransferase
SM757_09155	1206	JAXOJX010000011	3-oxoadipyl-CoA thiolase	H16_A1438	phaA	Acetyl-CoA acetyltransferase
SM757_01140	1176	JAXOJX010000001	thiolase	H16_A1438	phaA	Acetyl-CoA acetyltransferase
SM757_31645	1206	JAXOJX010000098	3-oxoadipyl-CoA thiolase	H16_A1438	phaA	Acetyl-CoA acetyltransferase
SM757_19540	1203	JAXOJX010000034	acetyl-CoA C-acyltransferase	H16_A1438	phaA	Acetyl-CoA acetyltransferase
SM757_22340	738	JAXOJX010000042 *	acetoacetyl-CoA reductase	H16_A1439; H16_A2002; H16_A2171	phaB1; phaB2; phaB3	Acetoacetyl-CoA reductase
SM757_08215	750	JAXOJX010000009	3-oxoacyl-ACP reductase FabG	H16_A1439; H16_A2002; H16_A2171	phaB1; phaB2; phaB3	Acetoacetyl-CoA reductase
SM757_31855	741	JAXOJX010000100	3-oxoacyl-ACP reductase FabG	H16_A1439; H16_A2002; H16_A2171	phaB1; phaB2; phaB3	Acetoacetyl-CoA reductase
SM757_23050	747	JAXOJX010000044	3-oxoacyl-ACP reductase FabG	H16_A2002; H16_A2171	phaB2; phaB3	Acetoacetyl-CoA reductase
SM757_08600	1344	JAXOJX010000010	polyhydroxyalkanoate depolymerase	H16_A1150; H16_A2862; H16_B0339; H16_B1014	phaZ1; phaZ2; phaZ3; phaZ5	intracellular poly(3-hydroxybutyrate
SM757_15980	1002	JAXOJX010000025	PHB depolymerase family esterase	H16_B2401	phaZ7	Poly(3-hydroxybutyrate) depolymeras
SM757_26695	2253	JAXOJX010000060	3-hydroxybutyrate oligomer hydrolase protein	H16_A2251	phaY1	D-(−)-3-hydroxybutyrate hydrolase
SM757_23895	870	JAXOJX010000047	alpha/beta hydrolase	H16_A1335	phaY2	D-(−)-3-hydroxybutyrate hydrolase
SM757_26850	558	JAXOJX010000062	phasin family protein	H16_A1381	phaP1	Phasin (PHA-granule associated protein)
SM757_25825	594	JAXOJX010000056	polyhydroxyalkanoate synthesis repressor PhaR	H16_A1440	phaR	transcriptional regulator

* *phaCAB* operon.

**Table 6 microorganisms-13-00137-t006:** Nucleotide identity among *phaC* from Choi et al. and *Azohydromonas phaC* genes.

Genus or Species (Strain)	Identity (%)	Alignment Length	Mismatches	E-Value
*australica* (DMS1124)	100.000	1611	0	0.0
Uncultured *Azohydromonas* sp.	94.727	1612	83	0.0
*aeria* (CFCC 13393)	91.646	1616	128	0.0
*lata* (NBRC102462)	90.627	1611	151	0.0
*lata* (H1)	90.503	1611	153	0.0
*caseinilytica* (G-1-1-1-4)	90.526	1615	145	0.0
*sediminis* (SYSU G00080)	80.455	1627	280	0.0
*Calidifontimicrobium* sp. (SYSU G02091)	80.086	1627	286	0.0

**Table 7 microorganisms-13-00137-t007:** Nucleotide identity among *phaA* from Choi et al. and *Azohydromonas phaA* genes.

Genus or Species (Strain)	Identity (%)	Alignment Length	Mismatches	E-Value
*australica* (DMS1124)	100.00	337	0	0.0
Uncultured *Azohydromonas* sp.	96.777	1179	38	0.0
*caseinilytica* (G-1-1-1-4)	96.438	1179	42	0.0
*lata* (NBRC 102462)	92.881	1180	82	0.0
*lata* (H1)	92.373	1180	88	0.0
*sediminis* (YIM 73032)	84.338	1187	170	0.0
*sediminis* (SYSU G00080)	84.233	1186	173	0.0
*Calidifontimicrobium* sp. (SYSU G02091)	83.825	1187	176	0.0
*aeria* (CFCC 13393)	80.086	128	16	0

**Table 8 microorganisms-13-00137-t008:** Nucleotide identity among *phaB* from Choi et al. and *Azohydromonas phaB* genes.

Genus or Species (Strain; Accession)	Identity (%)	Alignment Length	Mismatches	E-Value
*caseinilytica* (G-1-1-1-4)	94.851	738	38	0.0
*aeria* (CFCC13393; WP_157271469.1_7046)	94.580	738	40	0.0
*australica* (DSM 1124)	94.309	738	42	0.0
*aeria* (CFCC13393; WP_157272394.1_2294)	94.038	738	44	0.0
*lata* (H1)	93.496	738	48	0.0
*lata* (NBRC 102462)	93.360	738	49	0.0
*Calidifontimicrobium* sp. (SYSU G02091)	84.409	744	104	0.0
*sediminis* (SYSU G00080)	84.430	745	102	0.0
*sediminis* (YIM 73032)	84.161	745	104	0.0

## Data Availability

All sequence data are publicly available at NCBI under the GenBank accession number GCA_034427735.1. All data generated or analyzed during this study are included in Section 3 (Results) or in Appendix A of this paper. Scripts and auxiliary files generated for the current study are available at Zenodo in the repository at https://zenodo.org/records/13152277 (accessed date: 1 August 2024).

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
