# Peer review of "De Novo Assembly of the Polyhydroxybutyrate (PHB) Producer Azohydromonas lata Strain H1 Genome and Genomic Analysis of PHB Production Machinery"

_microorganisms, 2025, doi:10.3390/microorganisms13010137_

Round 1
Reviewer 1 Report
Comments and Suggestions for Authors
The manuscript describes the sequencing and genomic analysis of Azohydromonas lata strain H1. A. lata is a gram-negative bacterium that can produce PHB, a desired biopolymer. The authors, through there analysis of the A. lata geneome identified a large megaplasmid, and their main analysis focused on this megaplasmid. The authors note several distinctive features in the megaplasmid, including a series of transposes, toxin-antitoxin pairs, a CRISPR array, and several genes associated with PHB production including a phaCAB operon and several other Pha proteins that do not appear in an operon architecture. The authors assign appropriately classify the Pha proteins based on sequence similarities.
The methods are appropriately and adequately descripted, and the analysis is well based on the data presented. My main concerns are with the text and figures. For instance, in Figures 1, 2, the annotations are incredibly small and very difficult to read, as is Figure 6 (the portion of a sequence alignment). In the heat map figures (Figs. 3, 4, and 5) are slightly better, but the text is still quite small; the authors could have used different labels for readability and used the legends to describe the labels appropriately to make it much more readable and accessible for the reader. Additionally, the introduction and discussions are each a single mega-paragraph, akin to the megaplasmid the authors identified in this study. I strongly recommend that they carefully review and revise the text for readability by the audience.
Author Response
Thank you very much for taking the time to review this manuscript. Please find the detailed responses below and the corresponding revisions highlighted in the re-submitted file
Comment 1
….in Figures 1, 2, the annotations are incredibly small and very difficult to read, as is Figure 6 (the portion of a sequence alignment). In the heat map figures (Figs. 3, 4, and 5) are slightly better, but the text is still quite small; the authors could have used different labels for readability and used the legends to describe the labels appropriately to make it much more readable and accessible for the reader
Reply 1
Thank you for pointing this out. To make figures more readable we have revised all the annotations and, in the case of figures 3 and 4, accessions have been removed by referring to the Table S1 in the captions.
Comment 2
The introduction and discussions are each a single mega-paragraph, akin to the megaplasmid the authors identified in this study. I strongly recommend that they carefully review and revise the text for readability by the audience.
Reply 2
We appreciate the very useful suggestion. We have revised the text of introduction and discussion by dividing the entire text in smaller paragraphs. Additionally, more explicative sentences have been added and single words have been changed in the original text, to support the importance of studying PHB bacteria producers.
Reviewer 2 Report
Comments and Suggestions for Authors
1. I would recommend the authors to address an issue of different strains used to produce PHB for example Escherichia coli and others reported elsewhere doi.org/10.3390/polym13010108
2. In Figure there are a lot of small details which are barely seen. I would recommend to increase the fonts or optimize the content. I would also recommend to check other figures for consistency.
3. In general, the family of polyoxyalkanoates are rather big and there are a few representatives of this family such as PHBHV, and others, address an issue of the importance of PHB.
4. There is no conclusion section. I would recommend to provide some the most important comments/remarks in conclusion.
5. PHB production using different strain is very limited and does not meet the demands of industry. How this important point can be addressed to make application of PHB-based materials more widespread.
Author Response
Thank you very much for taking the time to review this manuscript. Please find the detailed responses below and the corresponding revisions highlighted in the re-submitted file
Comment 1
I would recommend the authors to address an issue of different strains used to produce PHB for example Escherichia coli and others reported elsewhere doi.org/10.3390/polym13010108
Reply 1
We thank the reviewer for the relevant suggestion. Considering the reference doi.org/10.3390/polym13010108indicated by the reviewer, we have added in the text citations of more Gram-negative genera usually considered for the PHB production in addition to recombinant E. coli, C. necator H16 and A. lata H1 (lanes 66-68).
Comment 2
In Figure there are a lot of small details which are barely seen. I would recommend to increase the fonts or optimize the content. I would also recommend to check other figures for consistency.
Reply 2
We agree with this comment. We have optimised the text embedded in all the figures by increasing the font size and, when possible, by removing information already present in supplementary tables (see figures 3 and 4 captions). See also reply 1 to the reviewer 1
Comment 3
In general, the family of polyoxyalkanoates are rather big and there are a few representatives of this family such as PHBHV, and others, address an issue of the importance of PHB.
Reply 3
We realised that the original manuscript does not emphasize enough the importance of PHB among the PHA family. Now we have better highlighted both in introduction (lanes 48-65) and discussion (lanes 413-420) the possible important role played by PHB, in the large PHA family, with respect to the growing demand of biodegradable bioplastics
Comment 4
There is no conclusion section. I would recommend to provide some the most important comments/remarks in conclusion.
Reply 4
Now the conclusion section has been added (lanes 469-483)
Comment 5
PHB production using different strain is very limited and does not meet the demands of industry. How this important point can be addressed to make application of PHB-based materials more widespread
Reply 5
This valuable comment has contributed to improve introduction, discussion and conclusion sections. In introduction (lanes 60-65), discussion (lanes 415-420) and conclusion (lanes 469-473) we have addressed the importance of studies, also by using omic approach, to improve the use of PHB mainly based on the production by low-cost substrates.